# Proton-assisted electron transfer and hydrogen-atom diffusion in a model system for photocatalytic hydrogen production

Yuanzheng Zhang [1], Yunrong Dai[2,3], Huihui Li [1], Lifeng Yin[1,3✉] & Michael R. Hoffmann[3✉]

Solar energy can be converted into chemical energy by photocatalytic water splitting to produce molecular hydrogen. Details of the photo-induced reaction mechanism occurring on the surface of a semiconductor are not fully understood, however. Herein, we employ a model photocatalytic system consisting of single atoms deposited on quantum dots that are anchored on to a primary photocatalyst to explore fundamental aspects of photolytic hydrogen generation. Single platinum atoms ($Pt_1$) are anchored onto carbon nitride quantum dots (CNQDs), which are loaded onto graphitic carbon nitride nanosheets (CNS), forming a $Pt_1@CNQDs/CNS$ composite. $Pt_1@CNQDs/CNS$ provides a well-defined photocatalytic system in which the electron and proton transfer processes that lead to the formation of hydrogen gas can be investigated. Results suggest that hydrogen bonding between hydrophilic surface groups of the CNQDs and interfacial water molecules facilitates both proton-assisted electron transfer and sorption/desorption pathways. Surface bound hydrogen atoms appear to diffuse from CNQDs surface sites to the deposited $Pt_1$ catalytic sites leading to higher hydrogen-atom fugacity surrounding each isolated $Pt_1$ site. We identify a pathway that allows for hydrogen-atom recombination into molecular hydrogen and eventually to hydrogen bubble evolution.

[1] State Key Laboratory of Water Environment Simulation, School of Environment, Beijing Normal University, Beijing, China. [2] School of Water Resources and Environment, China University of Geosciences (Beijing), Beijing, P. R. China. [3] Division of Engineering and Applied Science, Linde-Robinson Laboratory, California Institute of Technology, Pasadena, CA 91125, USA. ✉email: lfyin@bnu.edu.cn; mrh@caltech.edu

Over the past decade there has been surge in research into renewable and clean energy sources especially in light of concerns about climate change[1,2]. Hydrogen production via photocatalytic water splitting has been explored extensively[3,4]. Many groups are engaged in the development of new photocatalytic materials that are activated with broad spectrum UV–visible light available in incident solar radiation. $H_2$ bubble evolution at the semiconductor interfaces is preceded by (a) light absorption at wavelengths equal to or less than the bandgap energy, (b) excited hole and electron ($h^+/e^-$) formation and transport to the interface, (c) $h^+/e^-$ trapping at the solid–water interface, (d) proton transfer and diffusion at the interface, and $H_2$ production and desorption steps[5,6]. Strategies to improve the activity of various semiconductor photocatalysts include optimization of band structures to broaden light-absorption ranges and to enhance electron–hole separation efficiencies[7,8].

For another, proton transfer and proton-assisted electron transfer play a key role in determining specific reaction pathways and photocatalytic efficiencies[9–11], but there is no consensus on the mechanism of dynamic interfacial proton transfer. Previous studies have reported that molecularly adsorbed water or methanol can mediate the long-range proton transport on active photocatalytic surfaces[12–15]. It has been shown that phosphoric acid facilitates proton transfer and leads to a significant enhancement in the photocatalytic production of $H_2$[16–18]. In addition, nanomaterials that function as solid-state electrolytes (e.g., metal-organic framework compounds and graphene oxide) have been found to be efficient in proton transfer media, which have improved the performance of high-efficiency fuel cells[19–21]. Compared with liquid-phase proton carriers, solid proton conductors are able to transfer the protons while preserving their physical properties and spatial orientation[22,23]. A key research focus is to differentiate the proton transfer functions of solid-phase proton conductors from their activity as co-photocatalysts that facilitate photocatalytic $H_2$ production.

It is generally accepted that photocatalytic $H_2$ production pathways occur according to either the Volmer–Heyrovsky or Volmer–Tafel mechanism (Supplementary Note 1)[24]. In addition, photocatalytic $H_2$ production can be significantly enhanced by incorporating noble metals as co-catalysts[25]. However, the actual role of noble metal co-catalysts in the proton reduction reaction remains elusive. Metal co-catalysts (e.g., Pt, Pd, Ni, Cu) (i) trap electrons and facilitate the proton reduction reaction[25]; (ii) provide catalytic sites for the recombination of H-atoms[26]; and (iii) accelerate the formation of gaseous molecular hydrogen[27,28]. Furthermore, Pt-decorated materials are also thought to assist during interfacial proton transfer[12,29–31]. This is mainly due to the fact that not all atoms in the particle co-catalyst have catalytic activity, which limits to elucidate the relationship between the active center and their catalytic performance. With recent advances in the study of single-atom catalysts, the mechanistic understanding the nature of heterogeneous catalysis at an atomic level is improving[28,32].

In this study, we report on a photocatalytic system that consists of a photocatalyst, a proton carrier, and an active site for the hydrogen evolution. We have named the hydrogen evolution site the "active volcanic site", which facilitates gaseous hydrogen evolution in a manner similar to a volcanic eruption. Our hybrid photocatalyst is composed of single Pt atoms ($Pt_1$, the $H_2$ volcano-like evolution sites) deposited on to carbon nitride quantum dots (CNQDs). The CNQDs function as proton transporters, which are in turn uniformly loaded on to graphitic carbon nitride nanosheets resulting in mixed function $Pt_1$@CNQDs/CNS composite catalyst. Herein, we systematically investigate the relative roles of the CNQDs, CNS, and $Pt_1$ leading up photocatalytic hydrogen evolution.

## Results and discussion

**Model photocatalytic system preparation.** CNSs were prepared from urea at 600 °C in air for 4 h, while the CNQDs were synthesized from cyanuric chloride and melamine by a solvothermal method followed by moderate exfoliation as described in detail in the "Methods". The synthetic procedures for preparation of the $Pt_1$@CNQDs/CNS "volcanic island highways" are illustrated in Supplementary Fig. 1. $Pt_1$ were first deposited onto the surface of the CNQDs through a facile photochemical reduction method (denoted as $Pt_1$@CNQDs)[33,34]; this was followed by self-assembly of the $Pt_1$@CNQDs onto CNS. In this system, CNS is not only a source of photogenerated electrons but also a two-dimensional support for anchoring the CNQDs. The hydrophilic groups on the CNQDs serve as a hydrogen-bonded network with surface-adsorbed water molecules. Furthermore, the $Pt_1$ deposits, unlike Pt nanoparticles or nanoclusters (similar to the size of CNQDs and easily to conceal the role of CNQDs), provide numerous active sites on nano-sized CNQDs for the production of atomic hydrogen to readily diffuse to isolated $Pt_1$ sites to recombine and form $H_2$.

**Characterization of CNQDs/CNS.** The transmission electron microscopy (TEM) image (Fig. 1a and Supplementary Fig. 2) of the CNS shows a uniform layer-by-layer structure with a smooth external surface. The representative TEM images of the CNQDs (Fig. 1b and Supplementary Fig. 3) indicated that the synthesized CNQDs were uniform in size and well-distributed without aggregation. High-resolution (HRTEM) and aberration-corrected scanning TEM (AC-STEM) images of the CNQDs/CNS (Fig. 1c and Supplementary Fig. 4) show a typical lattice distance of 0.278 nm, which corresponds to the (200) facet of hexagonal $\beta$-$C_3N_4$ (JCPDS No. 87-1523)[35]. The particle size distribution was constrained between 3 and 4 nm (Supplementary Fig. 5).

UV–vis diffuse reflectance spectroscopy (DRS) and steady-state photoluminescence (PL) spectra were obtained to determine the optical properties of the CNQDs (Supplementary Figs. 6 and 7). DRS patterns show that the prepared samples absorb visible light at $\lambda < 460$ nm (Supplementary Fig. 8). The corresponding bandgap energies ($E_g$) were determined to be 2.80 eV for the CNS and 2.85 eV for the CNQDs/CNS based on *Tauc* plots (Fig. 1d)[36]. As shown in Fig. 1e, the valence band (VB) of the CNS and CNQDs/CNS was determined using ultraviolet photoelectron spectroscopy (UPS to be 5.90 and 6.46 eV vs. vacuum level by subtracting the width of the He I UPS spectra from the excitation energy (21.22 eV)[3]. Figure 1f displays the band structure diagram of the CNS and CNQDs/CNS. The VB and CB (conduction band) values expressed in electron volts can be converted to electrochemical energy potentials referenced to a reversible hydrogen electrode (RHE), which equals −4.44 eV vs. vacuum level[37]. The CB edge was thus determined to be −1.34 and −0.83 V (vs. RHE) for the CNS and CNQDs/CNS, respectively. Clearly, the CB edge of the composite CNQDs/CNS is substantially more negative than the standard state reduction potential for $H^+/H_2$ (0 V vs. RHE)[1]. Time-resolved PL spectra were used to explore the separation and recombination behaviors of photogenerated carriers in the CNQDs/CNS. The decay spectra of the CNS and CNQDs/CNS exhibit three radiative lifetimes with different contributions (Supplementary Fig. 9 and Supplementary Table 1). The average PL lifetime ($\tau_{av}$) is 5.98 ns for the CNS and 3.58 ns for the CNQDs/CNS. The decreased carrier lifetime indicates a lower quantity of quickly recombined carriers in the CNQDs/CNS[33]. The decreased PL emission spectra and increased photocurrent responses also confirmed that the recombination of photogenerated carriers was forcefully suppressed after loading the CNQDs (Supplementary Fig. 10)[5].

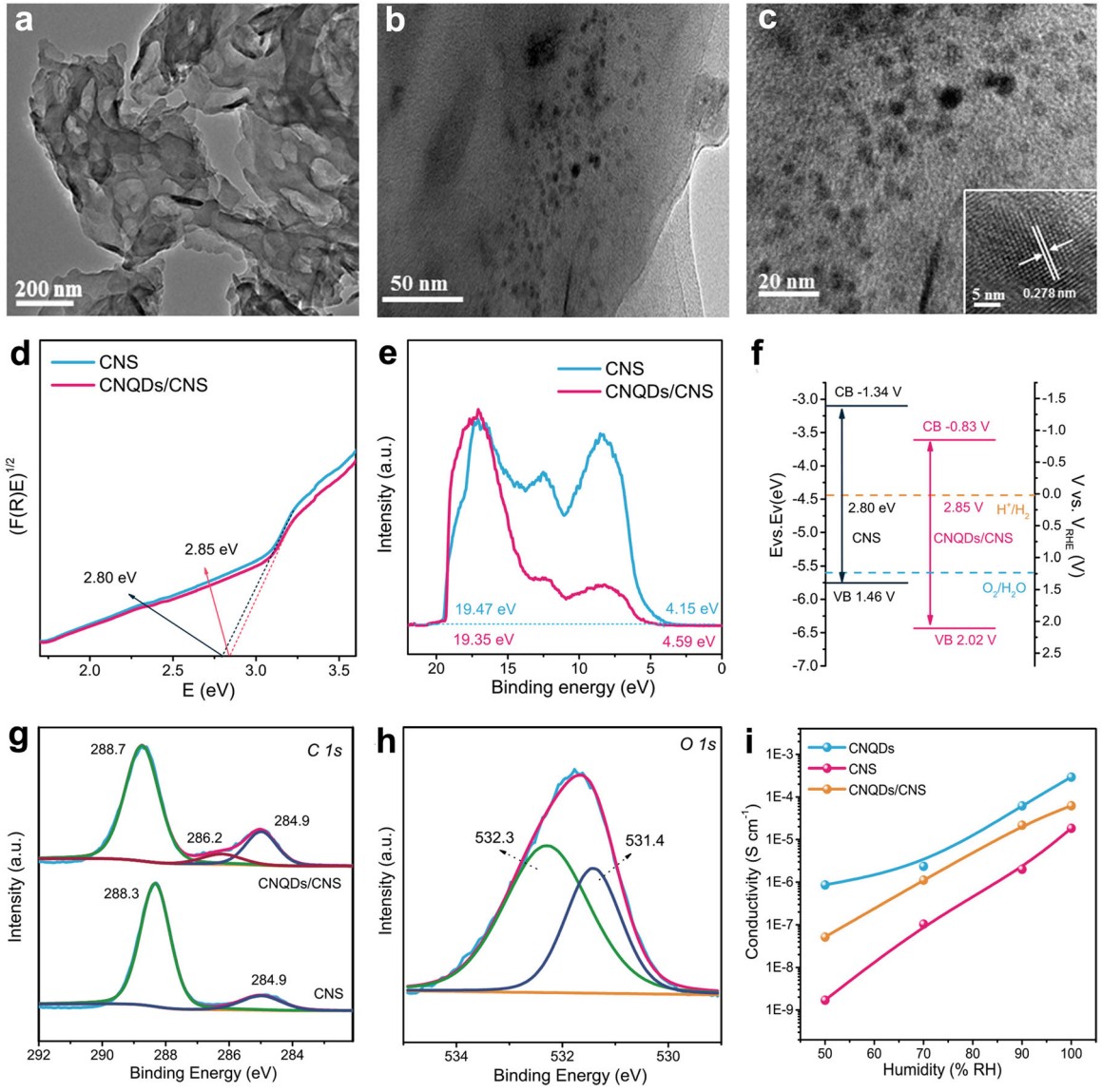

**Fig. 1 Structure, bandgap states, and proton conductivity of CNS and CNQDs/CNS. a** TEM image of as prepared CNS by chemically exfoliating graphitic carbon nitride. **b** TEM image of the CNQDs/CNS laminate structure. **c** HRTEM image of the CNQDs/CNS. Inset: the individual CNQDs embedded on the CNS. **d** Band gap measured by UV–vis diffuse reflectance spectroscopy (DRS). **e** UPS spectra vs. vacuum by subtracting the width of the He I UPS spectra from the excitation energy (21.22 eV). **f** The band structure diagram of the CNS and CNQDs/CNS. **g** High-resolution XPS spectra of C 1s of the CNS and CNQDs/CNS for C 1s energy level, **h** the O 1s spectra of the CNQDs/CNS. **i** Proton conductivities ($\sigma$) of the CNQDs, CNS, and CNQDs/CNS with respect to RH at 298 K.

The subtle changes of CNQDs/CNS on chemical states of the atoms were also observed on X-ray photoelectron spectroscopy (XPS). As shown in Supplementary Fig. 11, both samples are mainly composed of carbon, nitrogen, and oxygen elements, while a small amount of sulfur element was detected in CNQDs/CNS, which indicates the presence of sulfur-containing groups on the surface of the CNQD. The new S 2p peak of the CNQDs/CNS is located at 169.2 eV, corresponding to the –SO$_3$H functional group[38] (Supplementary Fig. 12), which was caused by the chemical oxidation treatment of CNQD by H$_2$SO$_4$. The C 1s peak (Fig. 1g, bottom) of the CNS is differentiated into two peaks at 288.3 and 284.9 eV, which are ascribed to the C–N=C and C–C groups of the CNS, respectively[39]. For the CNQDs/CNS (Fig. 1g, top), a C–O signal peak located at 286.2 eV appeared in the C 1s spectra[40]. The N 1s spectra for the CNQDs/CNS were deconvoluted into three peaks similar to that of the CNS (Supplementary Fig. 13). As shown in

Supplementary Fig. 14, small amounts of oxygen were detected in the CNS, which likely result from water absorption. While the CNQDs/CNS displayed two contributions at the binding energies of 531.4 and 532.3 eV, which are ascribed to O=S=O and C–O–H groups[41,42] (Fig. 1h), further indicating the effective protonation of CNQD in H$_2$SO$_4$ (ref. [42]). Moreover, the presence of –OH in the CNQDs/CNS is further confirmed by Fourier-transform infrared spectroscopy (FTIR), with a markedly enhanced vibration broad band[43] compared to that of the CNS at 3000–3500 cm$^{-1}$ (Supplementary Fig. 15). Based on the above results, we believe that the CNQDs can be uniformly loaded onto the CNS. The various surface hydrophilic functional groups on the CNQDs most likely facilitate proton transfer.

**Evidence for the role of CNQDs**. In order to evidence the CNQDs function as the proton transporter, the proton

conductivity tests of the CNQDs, CNS, and CNQDs/CNS were measured by alternating current (AC) impedance spectroscopy measurements in a reaction chamber with controlled humidity at room temperature[44]. Supplementary Figure 16 shows the Nyquist plots of the CNQDs, CNS, and CNQDs/CNS at 100% relative humidity (RH). We also obtained the AC impedance spectra of the samples at 50% RH to 100% RH at 298 K (Supplementary Figs. 17–19). The results show that the surface impedance of the samples decreases immediately with the increase of humidity. This clearly indicates that water molecules adsorbed in the surface of the CNQDs, CNS, and CNQDs/CNS plays an important role in the proton conduction pathway. Moreover, the CNQDs showed the minimum impedance value, which further evidence for the hydrophilic nature of the CNQDs is closely related to proton transport. As shown in Fig. 1i, the CNQDs modification indeed reduces the impedance of the CNS. The calculated conductivities ($\sigma$) for the CNQDs, CNS, and CNQDs/CNS at 100% RH at 298 K were $2.94 \times 10^{-4}$, $1.85 \times 10^{-5}$, and $6.25 \times 10^{-5}$ S cm$^{-1}$, respectively. The $\sigma$ value for CNQDs/CNS is similar to the value obtained for graphene oxide nanosheets[21]. The proton conductivity of the composite CNQDs/CNS is 3.4 times that of the CNS alone when the RH reaches 100%. This behavior suggests that the CNQDs on the surface of the CNS have numerous hydrophilic groups (i.e. –OH, and –SO$_3$H), which extend outward and assembled into a dense hydrogen-bonding network with the surface-adsorbed water molecules[21]. The surface –O−H⋯O complexes, the weakening of the –O−H chemical bond, and the strengthening of the –O⋯H hydrogen bond results in a low barrier for proton transfer via a *Grotthuss*-like mechanism (Supplementary Note 2)[13,44,45]. In addition, fast surface proton transfer should lower the barrier for the surface diffusion of H-atoms during the H$_2$ evolution[13].

**Characterization of Pt$_1$@CNQDs/CNS**. The HRTEM image of the Pt$_1$@CNQDs/CNS shows the CNQDs loaded onto the CNS with clearly distinctive lattice fringes (Fig. 2a). No Pt particles or clusters were observed in the HRTEM image, as well as powder X-ray diffraction (XRD) pattern (Supplementary Fig. 20) (i.e., no peaks were indexed to crystallographic Pt on the Pt$_1$@CNQDs/CNS). However, energy-dispersive X-ray spectroscopy mapping confirmed that Pt was distributed uniformly over the entire plane (Supplementary Fig. 21). These results imply that Pt$^0$ may exist in an atomically dispersed form. In addition, the AC HAADF-STEM image shows Pt$_1$ (bright dots in Fig. 2b) dispersed over the sample surface at an atomic level. The Pt$_1$ appear to be co-located on the CNQDs (relatively bright areas) rather than on the CNS (dark areas). A typical magnified AC HAADF-STEM image of individual CNQDs is shown to highlight the anchored atoms, wherein Pt$_1$ are clearly marked by red circles for close observation (Fig. 2c). The Pt 4$f$ core-level spectra of the Pt$_1$@CNQDs/CNS display two peak regions located at 72.8 (Pt 4$f_{7/2}$) and 76.1 eV (Pt 4$f_{5/2}$), respectively, which can be ascribed to signals of Pt$^{2+}$ (Supplementary Fig. 22)[46,47]. The Pt content in the whole complex is measured to be 0.65 wt% according to inductively coupled plasma optical emission spectrometry (ICP-OES) analysis. Meanwhile, Pt$_1$ were also anchored onto the CNS (Pt$_1$@CNS) using the similar photo-reduction procedure (Supplementary Fig. 23).

X-ray fine structure (XAFS) analysis of the Pt L$_3$-edge was obtained to gain more insight into the electronic structure and specific chemical environment of the Pt$_1$ catalytic sites. The related Fourier-transform extended X-ray absorption fine structure (FT-EXAFS) spectrum in R space for the Pt$_1$@CNQDs/CNS shows no detectable Pt–Pt bonds, ruling out the existence of Pt particles or clusters (Fig. 2d). Instead, only a single peak at

approximately 1.6 Å, which can be attributed to the coordination between Pt and light C/N elements, is clearly observed. Figure 2e shows the wavelet transform (WT) of the Pt L$_3$-edge EXAFS oscillations based on Morlet wavelets[48,49]. The WT maximum at approximately 4.8 Å$^{-1}$ arising from the Pt-C/N coordination is well-resolved at 1.0–3.0 Å for the Pt$_1$@CNQDs/CNS. In contrast, the intensity maxima of the Pt foil and H$_2$PtCl$_6$ references at approximately 8.95 and 6.03 Å$^{-1}$ were associated with Pt–Pt and Pt–Cl contributions, respectively. The results further confirmed that the Pt species in the Pt$_1$@CNQDs/CNS clearly appear to be a single atom or a small cluster. Moreover, the quantitative structural configuration of Pt$_1$ in the Pt$_1$@CNQDs/CNS was determined by least-squares EXAFS fitting (Fig. 2f and Supplementary Fig. 24). The coordination number (N) of Pt$_1$ was ~5 Pt–N bond length of 2.02 Å (Supplementary Table 2), suggesting that the Pt$_1$ were dispersed on the top of the five-membered rings of the CN network[50]. The fitting curves of the Pt foil and H$_2$PtCl$_6$ are displayed in Supplementary Figs. 25 and 26.

**Photocatalytic performance and stability**. The individual materials and the composite catalyst photocatalytic H$_2$ production rates were determined in a sealed gas circulation system under visible-light irradiation. In the present study, no appreciable hydrogen production was detected in the absence of either visible-light irradiation or photocatalyst, suggesting that hydrogen was produced by photocatalytic reactions. Figure 3 summarizes the H$_2$ production rates using the CNS, Pt$_1$@CNQDs, CNQDs/CNS, Pt$_1$@CNS, and Pt$_1$@CNQDs/CNS as photocatalysts. It is worth noting that a loading of 0.5 wt% CNQDs onto CNS enhanced the photocatalytic H$_2$ production rate (5326 $\mu$mol h$^{-1}$ g$^{-1}$) by a factor of 9.4 when compared to that of CNS (565 $\mu$mol h$^{-1}$ g$^{-1}$). This order of magnitude enhancement due to the addition of CNQDs most likely resulted from a reduction in the energetic barrier for surface proton transfer (Supplementary Fig. 27). The photocatalytic H$_2$ production is almost negligible for Pt$_1$@CNQDs, which suggests that Pt$_1$@CNQDs is inert for photocatalytic H$_2$ production but play a role of co-catalyst. In contrast, the H$_2$ production rate of the Pt$_1$@CNS increased to 7683 $\mu$mol h$^{-1}$ g$^{-1}$, which was approximately 13.6 times higher than that of the CNS. Even a more substantial increase in the H$_2$ production rate (e.g., 19372 $\mu$mol h$^{-1}$ g$^{-1}$) for the Pt$_1$@CNQDs/CNS composite was observed, which is much higher than that of g-C$_3$N$_4$ and TiO$_2$ modified by other co-catalysts (Supplementary Fig. 28). Furthermore, the Pt$_1$@CNQDs/CNS composite photocatalyst was reused over multiple cycles without a noticeable decrease in the H$_2$ production rate (Supplementary Fig. 29). Then the samples after photocatalytic reaction was characterized to further investigate its stability. As evidenced by XRD patterns (Supplementary Fig. 30), the crystallinity and structure of the samples have not changed. The AC HADDF-TEM images of Pt$_1$@CNQDs/CNS after reaction reveals that Pt species are still anchored on the CNQDs in the form of isolated atoms (Supplementary Fig. 31). Moreover, the XANES and EXAFS spectra of the Pt$_1$@CNQDs/CNS before and after cyclic photocatalytic reaction showed the same peak position with similar intensity (Supplementary Fig. 32). The above results demonstrate the good stability of Pt$_1$@CNQDs/CNS.

**Mechanism investigation of molecular hydrogen generation**. A key challenge is to explain the role Pt$_1$ sites in terms of their roles as focal point outlet leading to H$_2$ evolution. It is well known that Pt-group metals are catalysts for the adsorption and recombination of reactive hydrogen species leading to H$_2$ bubble evolution[27]. Therefore, an in situ electron paramagnetic resonance (EPR) experiment involving the capture of H-atom was carried out to

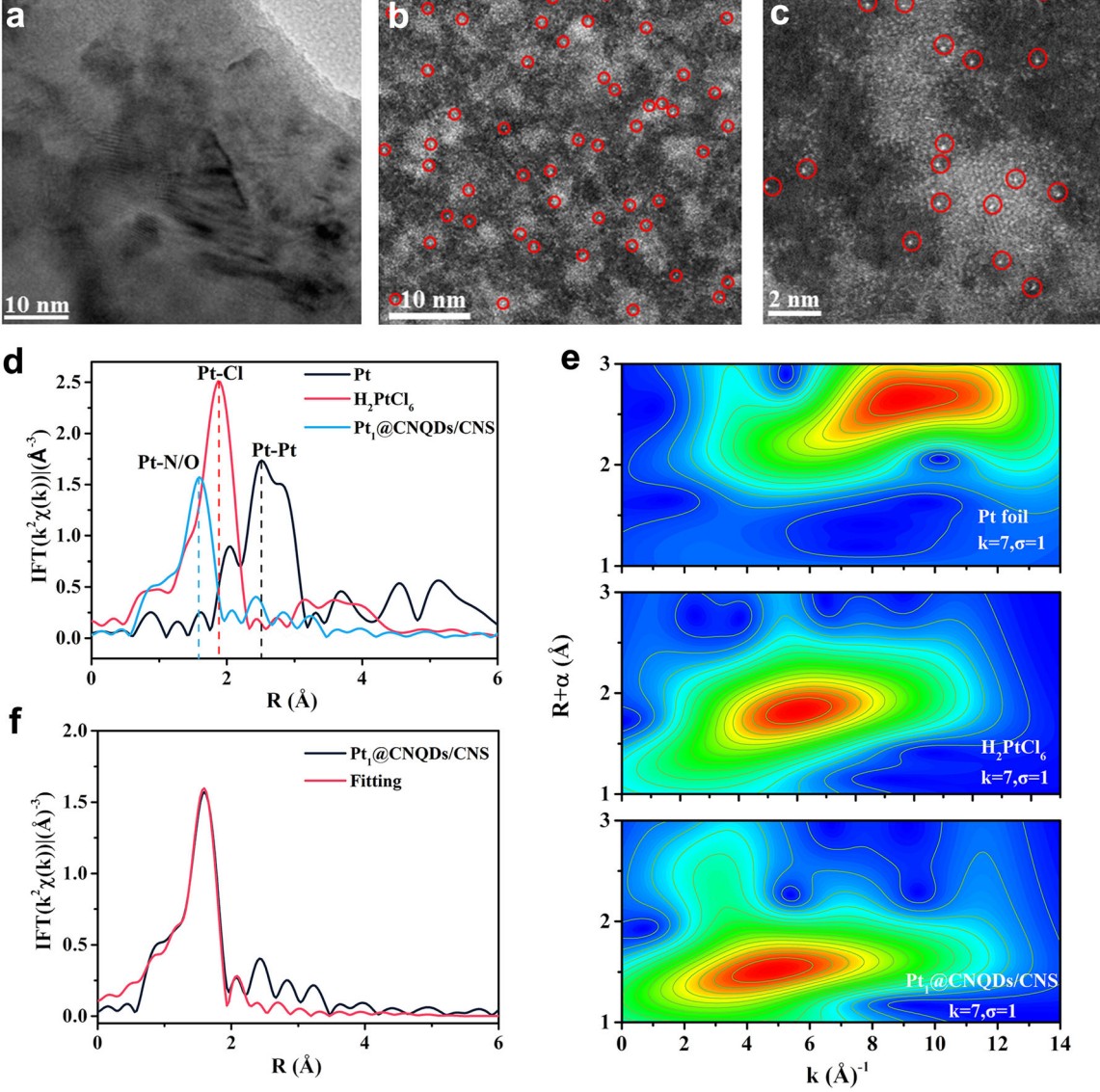

**Fig. 2 Structural characterizations of Pt$_1$@CNQDs/CNS.** **a** HRTEM image and **b**, **c** AC HAADF-STEM images of the Pt$_1$@CNQDs/CNS at the scale of 10 and 2 nm. **d** FT-EXAFS spectra and **e** WT-EXAFS spectra of the Pt L$_3$-edge of the Pt foil, H$_2$PtCl$_6$, and Pt$_1$@CNQDs/CNS. **f** The corresponding EXAFS R space fitting curves of the Pt$_1$@CNQDs/CNS.

investigate the intermediate state of hydrogen. Even though the transient H-atoms cannot be directly observed, the spin adducts formed by the addition of H-atom to 5,5-dimethyl-1-pyrroline-N-oxide (DMPO) can be analyzed by EPR[51]. As shown in Fig. 4a, a set of nine signal peaks (labeled as a solid rhombus) is observed in the spectrum after illumination for 20 s ($\lambda > 400$ nm), which is typical of a DMPO-H radical species[51,52]. In addition, a set of six signal peaks (labeled as a solid triangle) in the spectrum can be attributed to the spin adducts of C-center free radicals and DMPO[53]. It can be found that the signal of DMPO-H radical increases with the increase of illumination time, indicating the •H radical is an intermediate of the photocatalytic H$_2$ production from water (Supplementary Fig. 33). In order to better analyze the H-atoms transfer mechanism over different surfaces, the deconvolution of DMPO-H spectra was conducted using the Bruker Xepr software (Fig. 4b). In addition, the hyperfine coupling constants (hfcc) for nitrogen ($a_N$) and hydrogen ($a_H$) atoms are shown in Table 1, and the hfcc values are well consistent with reported values, further confirming the formation of •H

radical[54,55]. Moreover, the •H radical signal strengths of the samples were also compared (the peak area in Table 1). It can be seen that the signal strength of CNQDs/CNS is stronger than that of CNS, indicating that the CNQDs can boost proton reduction to H-atoms (H$^+$ + e$^-$ ⟶ •H). In contrast, the Pt$_1$@CNS results in a very low intensity of the H-atom, which is lower than that of the CNS. We propose that both the reduction of protons to H-atoms (H$^+$ + e$^-$ ⟶ •H) and the recombination of H-atoms to H$_2$ (2 •H ⟶ H$_2$) occur on the Pt$_1$ sites[56]. The signal strength of Pt$_1$@CNQDs/CNS is lower than that of the CNQDs/CNS because Pt$_1$ accelerates the H-atom recombination reaction (in aqueous solution 2 •H ⟶ H$_2$, $k = 1.2 \times 10^{10}$ M$^{-1}$ s$^{-1}$)[57], resulting in a decrease in the signal of H-atom. Based on the above analysis, it is reasonable to explain that the reduction of protons to H-atoms takes place in CNQDs in the Pt$_1$@CNQDs/CNS system, and then the H-atoms are transferred to Pt$_1$ sites leading to recombination to form H$_2$. These results demonstrate that the Pt$_1$ sites serve as the H$_2$ bubble evolution outlets. It should also be noted hydrogen-atom transfer from the CNQDs to Pt$_1$ leading to recombination to

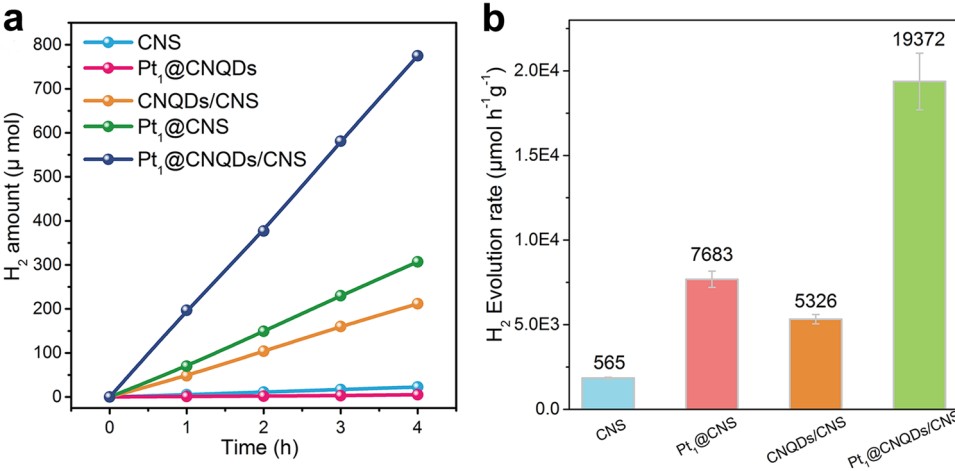

**Fig. 3 Photocatalytic H$_2$ production performance over the CNS-based catalysts. a** Time-dependent H$_2$ production. **b** Comparative H$_2$ production rates over different samples under visible light. The error bars represent standard deviations calculated from triplicate measurements. Reaction condition: 10 mg of the catalyst, 120 mL 10 vol% triethanolamine solution, under irradiation of a 300 W Xe lamp ($\lambda > 400$ nm).

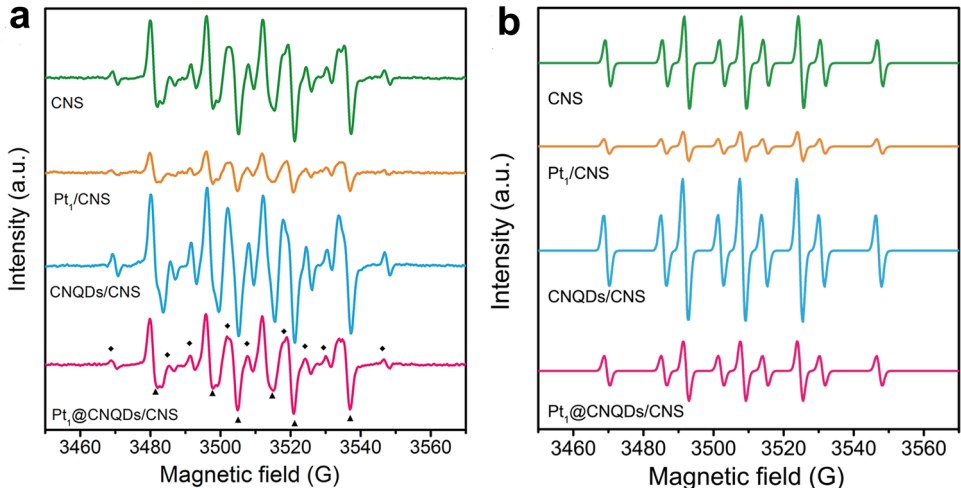

**Fig. 4 In situ EPR on intermediates. a** Experimental EPR signals of DMPO spin adducts generated under irradiation of a 300 W Xe lamp ($\lambda > 400$ nm) and **b** the deconvolution DMPO-H spectra according to the experimental signal in a diagram recorded for the CNS, Pt$_1$@CNS, CNQDs/CNS, and Pt$_1$@CNQDs/CNS.

**Table 1 Relevant data from simulations of the EPR spectra of DMPO-H radical.**

| Samples | $a_N$ (G) | $a_H$ (G) | NoH = $a_N/a_H$ | Peak area |
|---|---|---|---|---|
| CNS | 16.29 | 22.54 | 0.72 | 1.15 |
| Pt$_1$@CNS | 16.31 | 22.62 | 0.72 | 0.39 |
| CNQDs/CNS | 16.30 | 22.52 | 0.72 | 1.95 |
| Pt$_1$@ CNQDs/ CNS | 16.28 | 22.59 | 0.72 | 0.81 |

The hyperfine coupling constants were obtained by fitting spectra using Bruker Xepr software.

form H$_2$ is similar to the reverse hydrogen spillover phenomenon[58–60]. Although it is difficult to directly study the migration process of H-atoms, the spillover effect should influence the adsorption behavior of H$_2$ (ref. [61]). Thus, the H$_2$ uptake on different samples was examined by bubbling H$_2$ into the sample containing electrolyte solution at ambient temperature (Supplementary

Fig. 34). The results validate that the introduction of Pt$_1$ significantly increased the H$_2$ uptake capacity, which can be well explained by spillover effect[61]. Previous studies using isotope exchange recombination experiments have proven that the spillover effect is accompanied by reverse spillover[26,59]. On the other hand, theoretical calculations show that the metal sites on hydroxylated surfaces are expected to promote the reverse effect[62]. Therefore, it is definitely plausible that the Pt$_1$ sites on the functionalized CNQDs can induce a reverse spillover.

Based on the above experimental results, we suggest that a novel "volcanic islands highway" mechanism takes place leading to H$_2$ bubble evolution in the composite photocatalytic system, as shown in Fig. 5, UV–visible light irradiation of CNS (the "island") generates electrons that rapidly migrate to the surface of the CNQDs. The numerous inter-connected hydrophilic groups on the CNQDs provide a hydrogen-bonded network with adsorbed water, which then acts as a proton transfer channel (the "highway" component) that results in protons traveling on the "highway" to be reduced by the mobile conduction band electrons ($e^-_{cb}$) to form hydrogen atoms. Therefore, the CNQDs distributed on the

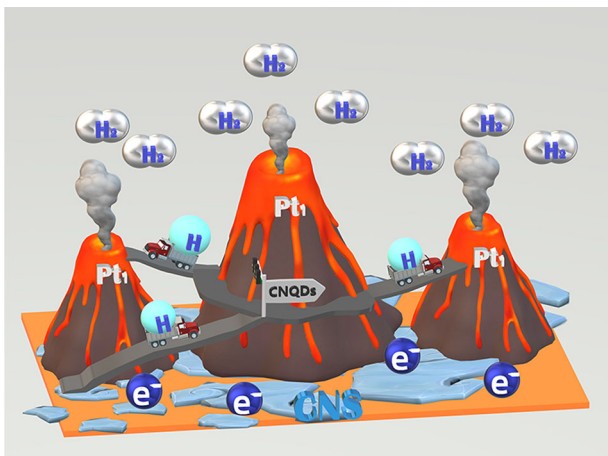

**Fig. 5 Schematic of the "volcanic islands highway" mechanism for photocatalytic H$_2$ production.** In the model system, CNS (the "island") worked as the source of photogenerated electrons and the support of Pt$_1$/CNQDs; CNQDs (the "highway") accelerate protons migration and make them combine with electrons to form H-atoms; while Pt$_1$ sites (the "volcanos") serve as the catalytic centers for H-atom recombination into molecular hydrogen.

CNS surfaces can not only act as proton-concentrating centers but also promote the $e^-_{cb}$ reduction of protons into H-atoms. Such correlated dynamics of electrons and protons are well demonstrated by similar surface functionalization strategy and has proven to be universally present in photocatalysts[11,63,64]. The H-atoms can be readily spilled over on Pt$_1$. This spillover zone around each Pt$_1$ are termed as "volcanos", in which the H-atoms recombine to form H$_2$ and eventually H$_2$-bubble evolution.

In summary, a Pt$_1$@CNQDs/CNS photocatalytic system with separated catalytic functions as been prepared. This composite catalyst allows for a mechanistic exploration of the fundamental processes leading photocatalytic H$_2$ evolution. A detailed array of experiments showed that the rate of H$_2$ production activity is, in part, controlled by the rate of interfacial proton transfer. The proton transfer pathways involve a hydrogen-bonded network involving CNQDs and adsorbed water molecules, while the Pt$_1$ sites serve as the dominate active catalytic centers that accelerate H-atom recombination into molecular hydrogen. A photocatalytic H$_2$ production model was used to determine key functional units leading to H$_2$ evolution processes. The key units identified include single-atom active centers and solid-state interfacial proton transfer pathways. Based on this model, a mechanism is proposed to explain the H$^+$–H•–H$_2$ sequential pathway involving an H$^+$–H• mass transfer step leading to an H$_2$ evolution outlet.

## Methods
**Synthesis of CNQDs and CNS.** CNQDs were fabricated by a facile low-temperature solvothermal method followed by moderate exfoliation[65]. The synthesis required mixing 7.0 mmol of cyanuric chloride and 3.5 mmol of melamine into 30 mL acetonitrile with stirring for 1 h. After the initial step, the solution was placed in a 100 mL Teflon-lined stainless-steel autoclave and heated at 180 °C for 48 h. The resulting precipitate was sequentially washed with ultrapure water and absolute ethanol several times and dried at 60 °C. In the next step, 10 mg of the produced power was added into 5 mL concentrated H$_2$SO$_4$ (98%). The resulting solution was then heated to 200 °C and maintained at that temperature for 20 min under continuous stirring. In the following synthetic step, the above mixture was poured into 300 mL ultrapure water and sonicated for 1 h (Uibra Cell VCX 105, SONICS, USA); the resulting suspension was a clear, light-yellow CNQDs solution. CNQDs (~0.03 mg mL$^{-1}$) were recovered by ultra-filtration-centrifugation (3K30, SIGMA, GER) for 0.5 h in a centrifugal tube (Amicon Ultra, 3KD).

CNS were prepared by high-frequency ultrasonic exfoliating bulk graphitic carbon nitride (bulk g-C$_3$N$_4$). Urea was placed in a covered quartz crucible (filling volume:

50%) and then heated at 600 °C in a muffle furnace programmed at a ramped heating rate of 4 °C/min for 4 h. After cooling to room temperature, the resultant light-yellow solid was milled into powder using an agate mortar. The ground bulk g-C$_3$N$_4$ powder was then dispersed in 50 mL of ultrapure water and subjected to high-frequency sonication (350 W, 800 kHz, Nanjing Aike) for 1 h at room temperature. The light-yellow CNS were dried in a vacuum oven at 60 °C for 12 h.

**Synthesis of CNQDs/CNS.** One hundred milligrams of CNS and a given amount of the CNQDs solution were mixed into 20 mL of ultrapure water and sonicated (200 W, 40 kHz) for 30 min to obtain a homogeneous suspension. The product suspension was then rapidly frozen with liquid nitrogen and then freeze-dried under a vacuum. The composite material was dried for 24 h. To explore the relative effect of the mass loading of CNQDs on the H$_2$ production, CNQDs-CNS with different mass ratios (0.1, 0.3, 0.5, 0.7, and 1%) were produced using the same procedures.

**Synthesis of Pt$_1$@CNQDs/CNS and Pt$_1$@CNS.** Pt$_1$ sites were deposited onto the CNQDs via photochemical reduction of platinum hexachloride. In all, 0.2 mL of H$_2$PtCl$_6$ solution (10 mg mL$^{-1}$) was added into 20 mL aqueous solution containing a given amount of the CNQDs in a stainless-steel photocatalytic reactor connected to a high-vacuum line system. After vacuum treatment (–0.2 Mpa) for 30 min, the reactor solution was irradiated using a 300 W Xe lamp with a 365 nm polarizing filter (8 mW cm$^{-2}$) for 1 h under continuous stirring to form Pt$_1$@CNQDs in solution. In the next step, 100 mg CNS was added into the Pt$_1$@CNQDs solution and sonicated (200 W, 40 kHz) for 30 min to obtain a homogeneous suspension. Then, the resulting product was isolated by vacuum freeze-drying. Pt$_1$@CNSs were also prepared by photo-deposition. One hundred milligrams of CNS were dispersed in 20 mL of deionized water and then 0.2 mL of H$_2$PtCl$_6$ solution (10 mg mL$^{-1}$) was added with sonication (200 W, 40 kHz) for 30 min. Subsequently, the mixture was irradiated using a 300 W Xe lamp with a 365 nm polarizing filter (8 mW cm$^{-2}$) for 1.5 h under continuous stirring. The resulting product was isolated by vacuum freeze-drying.

**Characterization.** TEM/high-resolution images were obtained using a JEOL 2100F microscope at an acceleration voltage of 200 kV. AC HAADF-STEM micrographs were obtained on the JEM-ARM 200F STEM fitted with a double aberration-corrector at 200 kV with a cold filed-emission gun. XPS spectra were analyzed to identify the surface chemical composition and electron structure, which was measured by an ESCSLAB 250Xi spectrometer with Al K$\alpha$-source radiation (1486.6 eV). FTIR spectra were recorded by a NEXUS 670 spectrometer to analyze the functional groups on the samples. UV–vis DRS spectra were acquired on a SPECORD 200 ultraviolet spectrophotometer. PL spectra were obtained on a FluoroMax-4 fluorescent spectrometer at room temperature and time-resolved PL spectra were conducted on a FluoroCube-TCSPC fluorescence lifetime spectrometer. The VB energy level of the samples was recorded on an ESCSLAB 250Xi spectrometer (UPS) equipped with a He I light source with the photon energy of 21.22 eV. ICP-OES was conducted using an axial view inductively coupled plasma spectrometer. EPR spectra of the hydrogen radicals were investigated using X band on a Bruker E500 spectrometer. The crystal structure of the as-synthesized products was characterized by XRD (X pert pro MPD) with Cu K$\alpha$ radiation at 40 kV, 40 mA.

**XAFS measurements and analysis.** The XAFS spectra at the Pt L$_3$-edge were measured at the BL14W1 beamline of the Shanghai Synchrotron Radiation Facility (SSRF). The data were collected in fluorescence mode using a Lytle detector in ambient conditions. All samples were pelletized as disks of 13 mm in diameter with 1 mm thickness by using LiF powder as the binder. Pt foil was used to calibrate the Si (111) double crystal monochromator. The acquired EXAFS data were processed and fitted with the ATHENA, ARTEMIS, and Hama-Fortran programs[66,67].

**Photocatalytic tests.** Photocatalytic H$_2$ evolution (HER) experiments were carried out in a top-down illumination reactor maid of stainless steel (200 mL) that was connected to a closed-loop gas system using a gas-circulated pump. In a typical photolysis experiment, 15 mg of the photocatalyst was dispersed in a 150 mL aqueous solution of 10 vol% triethanolamine (TEOA), which was used as an electron donor scavenger. Before irradiation, the suspension was exhausted for 10 min with a vacuum pump to remove dissolved air and purged with N$_2$ for 20 min. Subsequently, the reactor was exposed to a 300 W Xe lamp (HSX-F300) with a UV-cut filter (400 nm <$\lambda$< 780 nm). The suspension was stirred during the whole experiment. The produced gas was intermittently collected, and the amount of H$_2$ was analyzed by a gas chromatograph (Agilent 7890A) equipped with a thermal conductivity detector. The catalytic reaction temperature was maintained by a precise thermostat.

**Photo-electrochemical measurements.** The photocurrent was measured on a PGSTAT204 electrochemical analyzer (Autolab, Metrohm, NL) in a standard three-electrode configuration. The as-prepared samples were used as the working electrodes, while a Pt plate and a saturated calomel electrode were used as a counter electrode and a reference electrode, respectively. A 0.5 M Na$_2$SO$_4$ solution was used as the electrolyte. The light source was a 300 W Xe lamp equipped with an UV-cut filter.

The working electrodes were prepared by coating samples onto indium tin oxide (ITO) conductive glass. The ITO glass (2.0 cm × 2.0 cm) was pretreated in ultrapure water and ethanol for 10 min under ultrasonic conditions. Next, 5 mg samples were dispersed in a mixture of 1.5 mL of ultrapure water and isopropanol (volume ratio 1:2) and ultra-sonicated for 30 min to obtain a uniform slurry solution. Then, 80 μL of the slurry was dropped vertically on the surface of the ITO by micropipette and dried at room temperature. Finally, 20 μL of a Nafion solution was used to evenly cover the whole sample electrode, and the electrode was dried in air.

**Proton conductivity measurements**. AC impedance measurements were performed using a PGSTAT204 electrochemical analyzer (Autolab, Metrohm, NL) in an incubator at 298 K under various RH levels. The samples were intercepted on the nanofiltration membrane at a pressure of 1 MPa for 2 h, and then the membrane with dimensions of 1 cm ×1 cm was fixed horizontally between two platinum sheet electrodes (spacing is 0.2 cm) by insulated clamps. Impedance measurements were performed over a frequency range from 1 Hz to 1 MHz, and a 5 mV amplitude was used as the AC signal. The proton conductivities of the samples were calculated according to the following formula[68]:

$$\sigma = L/AR.$$

Here, $L$ is the platinum electrode spacing, $A$ is the area of the sample coverage between the electrodes, and $R$ is the resistance acquired from the Nyquist plots.

**$H_2$ uptake test**. The $H_2$ uptake experiments were used to verify the spillover effect. The tests were conducted in a 150 mL quartz glass flask connected to a closed-gas system with a gas-circulated pump. To evidence the hydrogen spillover effect in the photocatalytic process, the experiment was carried out under the same $H_2$ production aqueous environment. Typically, 200 mg of the samples was dispersed in 120 mL 10 vol% TEOA aqueous solution. The suspension was purged with $N_2$ for 30 min to remove dissolved air before reaction. Then, 300 μL of high-purity $H_2$ was injected into the sealed reactor, and the gas circulation pump was opened to allow the gas to continuously bubble into the solution. A 10 μL gas sample was periodically taken, and the $H_2$ concentration was measured by GC.

## Data availability
The authors declare that all the data supporting the findings of this study are available within the paper and its Supplementary Information.

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

## Acknowledgements

This work was financially supported by the National Natural Science Foundation of China (Grant 21777009), the Bill and Melinda Gates Foundation (BMGF RTTC Grants OPP1111246 and OPP1149755), Beijing Natural Science Foundation (Grant 8182031), and Major Science and Technology Program for Water Pollution Control and Treatment (Grant 2018ZX07109).

## Author contributions

Y.Z., Y.D., and H.L. prepared, characterized, and tested the catalysts. Y.D. performed the spectrum related experiments. Y.Z. and L.Y. collected and analyzed the data. Y.Z. and L.Y. wrote the paper in discussion with M.H. L.Y. and M.H. supervised the overall project.

## Competing interests

The authors declare no competing interests.
