## [Peer Review File · Communications Materials]

Web links to the author's journal account have been redacted from the decision letters as indicated to maintain confidentiality.

Decision letter and referee reports: first round

27th Feb 20

Dear Prof Yin,

Thank you for submitting your manuscript, "Photocatalytic hydrogen production on Pt₁@CNQDs/CNS composites: Proton-assisted electron transfer and H-atom diffusion", to Communications Materials. It has now been seen by 3 referees, whose comments are appended below. You will see that while they all find your work interesting, they have raised substantial concerns that must be addressed. In light of these comments, we cannot accept the manuscript for publication, but would be interested in considering a revised version that addresses these issues.

You will see in particular that Reviewer 3 does not feel that sufficient evidence is provided to support the conclusions. They say that the mechanistic role of single atom Pt and CNQDs has not been clarified, while they also raise concerns regarding the EPR data. Reviewer 1 and 2 also ask for some additional data.

We hope you will find the referees' comments useful as you decide how to proceed. Should further experimental data or analysis allow you to address these criticisms, we would be happy to look at a substantially revised manuscript. However, please bear in mind that we will be reluctant to approach the referees again in the absence of major revisions. If the revision process takes significantly longer than three months, we will be happy to reconsider your paper at a later date, as long as nothing similar has been accepted for publication at Communications Materials or published elsewhere in the meantime.

When submitting your revised manuscript, please include the following:

-A rebuttal letter with a point-by-point response to each of the referee comments and a description of changes made. Please include the complete referee report in the rebuttal letter. Please note that the rebuttal letter must be separate to the cover letter to the editors.

-A marked-up version of the manuscript with all changes to the text in red colored font. Please do not include tracked changes or comments. Please select the file type 'Revised Manuscript - Marked Up' when uploading the manuscript file to our online system.

-A clean version of the manuscript. Please select the file type 'Article File'.

-An updated <https://www.nature.com/documents/nr-editorial-policy-checklist.pdf> Editorial Policy checklist, uploaded as a 'Related Manuscript File' type. This checklist is to ensure your paper complies with all relevant editorial policies. If needed, please revise your manuscript in response to these points. Please note that this form is a dynamic 'smart pdf' and must therefore be downloaded and completed in Adobe Reader, instead of opening it in a web browser.

Please use the following link to submit your revised manuscript files:

[link redacted]

** This url links to your confidential home page and associated information about manuscripts you

may have submitted or be reviewing for us. If you wish to forward this email to co-authors, please delete the link to your homepage first **

Please do not hesitate to contact me if you have any questions or would like to discuss the required revisions further. Thank you for the opportunity to review your work.

Best regards,

John Plummer, PhD
Chief Editor
orcid.org/0000-0003-4824-8497
Communications Materials

Reviewers' comments:

Reviewer #1 (Remarks to the Author):

The author reported a Pt1 and CNQDs loaded carbon nitride photocatalyst for hydrogen evolution reaction and the mechanism was carefully investigated. The figures were properly organized to give a coherent story and the manuscript was well written. Most of the conclusions can be supported by the given data and evidence. However, the following concerns should be addressed before the manuscript is recommended to be published.

1. In XPS measurement, the intensity of the oxygen signal cannot reveal the content of oxygen in samples because most of the signal should come from the organic contamination in the environment (line 111-112). Solid evidence is needed to support the statement, for example, doing an elemental analysis testing.
2. Line 117, it is confusing that the author suddenly mentions O=S=O and C-O-H groups. If these groups are caused by the chemical oxidation treatment, the author should give a brief explanation in the context.
3. It is unpersuasive to claim the boost of proton reduction on CNQDs according to the increase of EPR signal (Figure 4b, line 208-209). If the author wants to insist, they should first prove that the signal is dose-related. For example, increasing the illumination time to check whether the signal also increases correspondingly.

Reviewer #2 (Remarks to the Author):

In this manuscript, authors report a hybrid photocatalyst composed of single platinum atoms anchored on to carbon nitride quantum dots, which in turn are loaded onto graphitic carbon nitride nanosheets. The Pt1@CNQDs/CNS provides a well-defined photocatalytic system in which the various electron and proton transfer steps leading to the formation of H₂ can be investigated using proton conductivity measurements, H₂ sorption/desorption measurements, and in situ electron paramagnetic resonance (EPR) free radical detection. The collective results suggest that hydrogen bonding between hydrophilic surface groups of the CNQDs and interfacial water molecules facilitates both proton-assisted electron transfer and sorption/desorption pathways. Surface bound hydrogen atoms appear to diffuse from CNQDs surface sites to the deposited Pt1 catalytic site leading to higher H-atom fugacity surrounding each isolated Pt1 island. "volcanic-island like flow pathway" that allows for H-atom recombination into molecular hydrogen and eventually to H₂ bubble evolution has been identified. This research topic is of great importance in optimizing molecular design and developing next-generation high-performance photocatalysts. Minor revision is needed before its publication.

1. The photocatalytic performance of the benchmark catalysts should be provided for a

comprehensive comparison.

2. What about the stability of the as-prepared catalyst?
3. Figure 3a, please add the photocatalytic performance of the Pt@CNQDs to illustrate the role of CNS.
4. Characterizations of the catalyst after catalysis should also be conducted, such as XRD, XPS, TEM...

Reviewer #3 (Remarks to the Author):

In my opinion, the idea of this work is good, which focus on the proton transfer process during photocatalytic hydrogen evolution. This is often ignored in many research papers. The conclusion is also interesting, claiming that CNQDs function as the solid-state proton transporters. However, one big problem in this work is that there's no strong evidence to support the conclusion. Therefore, I do not recommend the publication of this paper in current form. Several suggestions are listed below:

1. The author emphasis that Pt is presented in the system as single atom and adopted a series of advanced technologies to prove it; however, what is the advantage of single atom Pt in the present system? Is it indispensable for the claimed mechanism? Mechanism studies show no clue on it.
2. CNQDs could play many other roles instead of proton transporter in the present system, like photosensitizers and cocatalysts. Either of which could promote the efficiency of the system. How could the author confirm that it is the proton transporter plays the dominant role?
3. To prove the main opinion in the present system, EPR test was presented. Nevertheless, only EPR test and relevant tests in the supplementary data were adopted to support the mechanism. The quality of the signal is poor, especially when the quantitative comparison of signal intensity among different samples is considered; moreover, the explanation related to EPR test, which is vital to the conclusion of the paper, is weak.

Response to reviewer's comments

Response to reviewer 1

The author reported a Pt₁ and CNQDs loaded carbon nitride photocatalyst for hydrogen evolution reaction and the mechanism was carefully investigated. The figures were properly organized to give a coherent story and the manuscript was well written. Most of the conclusions can be supported by the given data and evidence. However, the following concerns should be addressed before the manuscript is recommended to be published.

√ Thanks for the reviewer's valuable comments and thoughtful suggestions on our manuscript. Based on these comments and suggestions, we have made careful modification on the original manuscript point by point to make the descriptions more clearly.

- 1 In XPS measurement, the intensity of the oxygen signal cannot reveal the content of oxygen in samples because most of the signal should come from the organic contamination in the environment (line 111-112). Solid evidence is needed to support the statement, for example, doing an elemental analysis testing.

√ We appreciate the reviewer's professional comments for the manuscript. We agree that the XPS results might be affected by the environmental contamination, and might fail to accurately reflect the content of each element in the sample, considering this we revised the related description for the XPS results (line 111-115). "As shown in Supplementary Fig. 11, both samples are mainly composed of carbon, nitrogen and oxygen elements, while a small amount of sulfur element was detected in CNQDs/CNS, which indicates the presence of sulfur-containing groups on the surface of the CNQD."

Supplementary Figure 11. XPS survey spectra of the CNS and CNQDs/CNS.

On the other hand, in this manuscript, our goal is not to measure the change of oxygen content in the samples, but to examine the evolution of oxygen-containing species on the surficial layer of samples. Otherwise, we believe the samples are all **absolutely** clean, sealed in a vacuum drier, from as-preparation to the XPS characterization. Herein, we supplement the CNS O 1s spectrum (Supplementary Figure 14) obtained in our previous work. It can be found that there is only one peak, however, there are two peaks showed in the patterns of CNQDs/CNS (Fig. 1h), which conveys that the CNQDs, at least, contains different oxygen-containing components.

In addition, we used energy dispersive X-ray spectroscopy (EDS) to test the elements of the samples, but there was no significant difference in the oxygen content.

Supplementary Figure 14. High-resolution X-ray photoelectron spectroscopy (XPS) spectra of O 1s for the CNS.

- 2 *Line 117, it is confusing that the author suddenly mentions O=S=O and C-O-H groups. If these groups are caused by the chemical oxidation treatment, the author should give a brief explanation in the context.*

√ Thanks a lot for the careful review and valuable comments. According to the reviewer's suggestion, we revised the manuscript. Based on the XPS and FT-IR analyses, we believe the surface functional groups were introduced during the process of protonation of CNQD in H₂SO₄. (line 111–115; line 118–121). The detailed descriptions are as follows:

“As shown in Supplementary Fig. 11, both samples are mainly composed of carbon, nitrogen and oxygen elements, while a small amount of sulfur element was detected in CNQDs/CNS, which indicates the presence of sulfur-containing groups on the surface of the CNQD. The new S 2p peak of the CNQDs/CNS is located at 169.2 eV, corresponding to the –SO₃H functional group³⁸ (Supplementary Fig. 12), which was caused the chemical oxidation treatment of CNQD by H₂SO₄. ...As shown in Supplementary Fig. 14, small amounts of oxygen were detected in the CNS, which likely result from water absorption. While the CNQDs/CNS displayed two contributions at the binding energies of 531.4 eV and 532.3 eV, which are ascribed to O=S=O and C–O–H groups^{41,42} (Fig. 1h), further indicating the effective protonation of CNQD in H₂SO₄.⁴²”

- 3 *It is unpersuasive to claim the boost of proton reduction on CNQDs according to the increase of EPR signal (Figure 4b, line 208-209). If the author wants to insist, they should first prove that the signal is dose-related. For example, increasing the illumination time to check whether the signal also increases correspondingly.*

√ Thank you for the suggestion. It's very important and quite helpful for our manuscript. In order to determine the reliability of signal intensity, we supplemented the change of signal intensity of the sample with the time of light irradiation (line 216–219). As shown in Supplementary Figure 33, the experimental results show that

the signal of H• radical increases with the increase of illumination time, indicating that the H• radical is an intermediate of the photocatalytic H₂ production from water. Thank you again for the valuable suggestion.

Supplementary Figure 33. a EPR spectra of DMPO adducts recorded for CNQDs/CNS before and after visible light irradiation ($\lambda > 400$ nm). b-d The amplification diagrams of the characteristic peaks of H• radical in figure a.

Response to reviewer 2

Comments to the Author

In this manuscript, authors report a hybrid photocatalyst composed of single platinum atoms anchored on to carbon nitride quantum dots, which in turn are loaded onto graphitic carbon nitride nanosheets. The $Pt_1@CNQDs/CNS$ provides a well-defined photocatalytic system in which the various electron and proton transfer steps leading to the formation of H_2 can be investigated using proton conductivity measurements, H_2 sorption/desorption measurements, and in situ electron paramagnetic resonance (EPR) free radical detection. The collective results suggest that hydrogen bonding between hydrophilic surface groups of the CNQDs and interfacial water molecules facilitates both proton-assisted electron transfer and sorption/desorption pathways. Surface bound hydrogen atoms appear to diffuse from CNQDs surface sites to the deposited Pt_1 catalytic site leading to higher H-atom fugacity surrounding each isolated Pt_1 island. volcanic-island like flow pathway that allows for H-atom recombination into molecular hydrogen and eventually to H_2 bubble evolution has been identified. This research topic is of great importance in optimizing molecular design and developing next-generation high-performance photocatalysts. Minor revision is needed before its publication.

√ Many thanks for the reviewer's advice. Based on these comments and suggestions, we have made careful modification on the original manuscript point by point to make the descriptions more clearly.

- 1 The photocatalytic performance of the benchmark catalysts should be provided for a comprehensive comparison.

√ Thanks for the referee's suggestion. In order to provide a comprehensive comparison of the photocatalytic properties of the samples, we tested the photocatalytic hydrogen production activity of some catalysts (as reference photocatalysts), including bulk $g-C_3N_4$, $NiCo(PO_4)_3/g-C_3N_4$, Pt Nanoparticles (Pt NPs)/ $g-C_3N_4$, and Pt NPs/defect TiO_2 . These photocatalysts have been studied by us and have good photocatalytic hydrogen production performance. However, in this study, the results show that the hydrogen production performance of the $Pt_1@CNQDs/CNS$ in the manuscript is much better than that of the reference photocatalysts (Supplementary Fig. 28).

Supplementary Figure 28. Photocatalytic activity of the samples bulk $g-C_3N_4$, $NiCo(PO_4)_3/g-C_3N_4$, Pt Nanoparticles (Pt NPs)/ $g-C_3N_4$, Pt NPs/defect TiO_2 , and $Pt_1@CNQDs/CNS$ for the photocatalytic H_2 production

under visible-light irradiation.

2 *What about the stability of the as-prepared catalyst?*

√ Thanks for the reviewer's valuable comments and thoughtful suggestions on our manuscript. We have investigated the stability of the best-performing Pt₁@CNQDs/CNS composite by repeatedly photocatalytic hydrogen production over four cycles under the same conditions (Supplementary Fig. 29).

Supplementary Figure 29. Recycling photoactivity tests of the Pt₁@CNQDs/CNS.

The Pt₁@CNQDs/CNS composite photocatalyst was reused over 4 cycles, 16 h, without a noticeable decrease in the H₂ production rate, indicating the good stability of the photocatalytic H₂ production system.

3 *Figure 3a, please add the photocatalytic performance of the Pt₁@CNQDs to illustrate the role of CNS.*

√ Thanks for the reviewer's valuable comments. According to the reviewer's suggestion, we added the photocatalytic activity of Pt₁@CNQDs, as shown in the modified Fig. 3a. As can be seen, the photocatalytic H₂ production is almost negligible for Pt₁@CNQDs, indicating that Pt₁@CNQDs is inert for photocatalytic H₂ production but play a role of co-catalyst and CNS as the main photocatalyst is essential.

Fig. 3 a H₂ production kinetics over different composite photocatalysts under visible light irradiation.

4 *Characterizations of the catalyst after catalysis should also be conducted, such as XRD, XPS, TEM*

√ Many thanks for the reviewer's advice. According to the reviewer's suggestion, we characterized the Pt₁@CNQDs/CNS after photocatalytic reaction, including XRD, AC HADDF-TEM and XAS spectra, and compared them with the fresh samples before the reaction to investigate the stability of the catalyst (Supplementary Fig. 30-32).

“As evidenced by XRD patterns (Supplementary Fig. 30), the crystallinity and structure of the samples have not changed. The AC HADDF-TEM images of Pt₁@CNQDs/CNS after reaction reveals that Pt species are still anchored on the CNQDs in the form of isolated atoms (Supplementary Fig. 31). Moreover, the XANES and EXAFS spectra of the Pt₁@CNQDs/CNS before and after cyclic photocatalytic reaction showed the same peak position with similar intensity (Supplementary Fig. 32). The above results demonstrate the good stability of Pt₁@CNQDs/CNS.” (line 194–200)

Supplementary Figure 30. The XRD patterns of Pt₁@CNQDs/CNS before and after cyclic photocatalytic reaction.

Supplementary Figure 31. a,b AC HAADF-STEM images of the Pt₁@CNQDs/CNS after cyclic photocatalytic reaction.

Supplementary Figure 32. a The Pt L₃-edge XANES spectra and **b** FT-EXAFS spectra of the Pt₁@CNQDs/CNS before and after cyclic photocatalytic reaction.

Response to reviewer 3

Comments to the Author

In my opinion, the idea of this work is good, which focus on the proton transfer process during photocatalytic hydrogen evolution. This is often ignored in many research papers. The conclusion is also interesting, claiming that CNQDs function as the solid-state proton transporters. However, one big problem in this work is that there is no strong evidence to support the conclusion. Therefore, I do not recommend the publication of this paper in current form. Several suggestions are listed below:

√ We appreciate the reviewer's professional comments and agree the suggestion from reviewers. Based on these comments and suggestions, we have made modification on the original manuscript point by point as follows.

- 1 *The author emphasis that Pt is presented in the system as single atom and adopted a series of advanced technologies to prove it; however, what is the advantage of single atom Pt in the present system? Is it indispensable for the claimed mechanism? Mechanism studies show no clue on it.*

√ Thanks for the reviewer's valuable comments and thoughtful suggestions on our manuscript. We believe there is completely different between bulk Pt and single atomic Pt on their catalytic actions, which might bring about the distinct reaction pathways of hydrogen over them. We complement these viewpoints to our revised manuscript and the detailed description can be found in line 51-54, Page 2 and line 71-73, Page 3.

It is well known that Pt nanoparticles usually act as the active center for photocatalytic hydrogen evolution reaction. However, there are still some speculations regarding the actual role of the noble metal co-catalyst. This is largely due to the fact that not all the atoms in particulate catalysts are catalytically active, which limits to elucidate the relationship between the active center and their catalytic performance. (line 51–54). In the case of Pt nanoparticles, they are composed of tons of Pt atoms and constructed by Pt–Pt bonds. On the Pt nanoparticle surface, H atoms can locate at uncertain Pt sites, making it difficult to elucidate the formation mechanism of hydrogen molecules. However, the uniform and isolated dispersed Pt (single) atoms with clear active sites provide a powerful tool for investigating the relationship between the single atom structure and their catalytic performance. Therefore, a unique photocatalytic hydrogen evolution reaction pathway involving H-atom diffusion is reported based on Pt₁@CNQDs/CNS.

Furthermore, the multi-scale well-defined photocatalytic hybrid system consisting of single atom (Pt₁)-nanometer (CNQDs)-micron (CNS) was also conducive to revealing the roles of each component in photocatalytic reaction. The dispersed Pt single atoms, unlike Pt nanoparticles or nanoclusters (similar to the size of CNQDs and easily to conceal the role of CNQDs), provide numerous active sites on nano-sized CNQDs for the production of atomic hydrogen to readily diffuse to isolated Pt₁ sites to recombine and form H₂ (line 71–73).

- 2 *CNQDs could play many other roles instead of proton transporter in the present system, like photosensitizers and cocatalysts. Either of which could promote the efficiency of the system. How could the author confirm that*

it is the proton transporter plays the dominant role?

√ Much thanks for your professional comments on our work. Just like what the reviewer mentioned, quantum dots (QDs) has been used as photosensitizers to promote the absorption of light and as a co-catalyst to promote the separation of photo-generated electron-hole pairs. In our system, CNQDs do not show obvious photosensitization as shown in the Supplementary Figure 8. However, our results show that CNQDs indeed act as the center of electron absorption promote the migration of photo-generated electrons from CNS to CNQDs to improve the photocatalytic efficiency. It should be noted that the interfacial redox reactions of photocatalysts often include the simultaneous gain of electrons and protons not just in terms of electron transfer, they are also affected by proton activity. (Science, 2012, 336, 1298–1301; ChemCatChem 2015, 7, 724–731). Therefore, we believe that the activities of electrons and protons on the CNQDs (with hydrophilic groups: hydroxyls, sulfonic acid) are closely related and mutually complementary. On the one hand, CNQDs promote photo-generated electrons to migrate from CNS to itself; on the other hand, hydrophilic groups on the surface of CNQDs promote proton adsorption and transport, in which the adsorbed protons are reduced to hydrogen atoms by electrons. Therefore, the CNQDs both facilitate the fast transfer of proton and electron and promote the reduction of protons into H-atoms. Such correlated dynamics of electrons and protons are well demonstrated by similar surface functionalization strategy (Nano Energy, 2018, 50 383–392; Energy Environ. Sci., 2013, 6, 3665–3675) and are also verified to be universally present in photocatalysts (ACS Catalysis, 2017, 7, 2744-2752). (Line 247–250, page 11)

- 3 *To prove the main opinion in the present system, EPR test was presented. Nevertheless, only EPR test and relevant tests in the supplementary data were adopted to support the mechanism. The quality of the signal is poor, especially when the quantitative comparison of signal intensity among different samples is considered; moreover, the explanation related to EPR test, which is vital to the conclusion of the paper, is weak.*

√ Thanks a lot for the reviewer's in-depth analysis and insightful comments on our manuscript. Identification of the H^+ -H•- H_2 dynamic migration and transformation in the **actual catalytic** process is experimentally challenging. EPR spectroscopy that allows to directly observe and quantify H• radical intermediate in the photocatalytic systems under visible light irradiation and provides an approach for studying the mechanism. In order to improve the persuasion of EPR signal, we added the change of EPR signal intensity with light time (Supplementary Figure 33), which proves that the signal is dose-related. In order to further verify the diffusion behavior of hydrogen atom, we carried out the hydrogen adsorption experiment also under the **actual photocatalytic reaction conditions**, which demonstrates that the Pt₁ serve as the site of H-atoms recombination to form H₂. The data presented in this study are in line with observations from other article available in the literature (PNAS, 2014, 111, 7942-7947). As the reviewer said, it may be more persuasive to provide other experimental tests. However, it is a great challenge to detect the migration and transformation of hydrogen species in the real catalytic reaction conditions. We are very grateful to the reviewers for pointing out the limitations of our work. We will focus on this in the future research, and try to use H-D isotope exchange and ultra-high sensitivity in-situ Raman or infrared spectroscopy to further study at the molecular level.

Supplementary Figure 33. a EPR spectra of DMPO adducts recorded for CNQDs/CNS before and after visible light irradiation ($\lambda > 400$ nm). b-d The amplification diagrams of the characteristic peaks of $\text{H}\cdot$ radical in figure a.

Decision letter and referee reports: second round

18th May 20

Dear Prof Yin,

Thank you for submitting your revised manuscript, "Photocatalytic hydrogen production on Pt₁@CNQDs/CNS composites: Proton-assisted electron transfer and H-atom diffusion", to Communications Materials. It has now been seen again by the 3 referees, whose comments are appended below.

While Reviewer 1 and 2 are now supportive of publication, Reviewer 3 raises substantial concerns regarding experimental support for claims made in the paper. In light of these comments, we cannot accept the manuscript for publication, but are interested in considering a revised version that addresses these serious issues, by the provision of new data.

However, please bear in mind that we will be reluctant to approach the referees again in the absence of a convincing reply to these points. If the revision process takes significantly longer than three months, we will be happy to reconsider your paper at a later date, as long as nothing similar has been accepted for publication at Communications Materials or published elsewhere in the meantime.

When submitting your revised manuscript, please include the following:

-A rebuttal letter with a point-by-point response to each of the referee comments and a description of changes made. Please include the complete referee report in the rebuttal letter. Please note that the rebuttal letter must be separate to the cover letter to the editors.

-A marked-up version of the manuscript with all changes to the text in red colored font. Please do not include tracked changes or comments. Please select the file type 'Revised Manuscript - Marked Up' when uploading the manuscript file to our online system.

-A clean version of the manuscript. Please select the file type 'Article File'.

-An updated <https://www.nature.com/documents/nr-editorial-policy-checklist.zip> Editorial Policy checklist, uploaded as a 'Related Manuscript File' type. This checklist is to ensure your paper complies with all relevant editorial policies. If needed, please revise your manuscript in response to these points. Please note that this form is a dynamic 'smart pdf' and must therefore be downloaded and completed in Adobe Reader. Clicking this link will download a zip file containing the pdf.

Please use the following link to submit your revised manuscript files:

[link redacted]

We understand that due to the current global situation, the time required for revision may be longer than usual. We would appreciate it if you could keep us informed about an estimated timescale for resubmission, to facilitate our planning. Of course, if you are unable to estimate, we

are happy to accommodate necessary extensions nevertheless.

Please do not hesitate to contact me if you have any questions or would like to discuss the required revisions further. Thank you for the opportunity to review your work.

Best regards,

John Plummer, PhD
Chief Editor
orcid.org/0000-0003-4824-8497
Communications Materials

Reviewers' comments:

Reviewer #1 (Remarks to the Author):

The authors have fully addressed the concerns of this reviewer. I recommend the manuscript to be published in the present vision.

Reviewer #2 (Remarks to the Author):

The authors have satisfied me with respect to the points I raised previously.

Reviewer #3 (Remarks to the Author):

I am OK with the previous two comments. However, I insist that EPR data should be thoroughly reconsidered in this paper, which is directly related to the core conclusion of the work. Here are my concerns:

1. The author think that CNQDs is the proton transporter of the present system, what is the direct experiment or theoretical evidence to support this conclusion? Is it from EPR that CNQDs/CN has a stronger H radical signal than CN? If it is, why a strong H radical signal could be relevant to that CNQDs promote proton transport. If it is not, please give more detail that why CNQDs function as the proton transporter.
2. I cannot understand with the data presented Figure 4b, how could the author come to the description that "In sharp contrast to the CNS, the CNQDs/CNS produces a significantly enhanced H-atom signal peak" and "The Pt1@CNQDs/CNS composite has a comparable H-atom signal peak to that of the CNQDs/CNS."
3. From my observation, the intensity of Pt1@CNQDs/CNS has decreased obviously in contrast to CNQDs/CNS as well, which is similar to that of Pt1@CNS and CNS. This could be indicative that protons reduction still mainly occurs on Pt (conclusion obtained based on the author's analysis of Pt1@CNS and CNS), instead of CNQDs (which is claimed by the author).
4. Moreover, even if the above description is right and EPR test has been carried out to prove that the intensity of the signal is dose-related, I insist that these discrepancies could be a result of accidental error because they are so small. Repeated EPR should be carried out to prove the data reliability.

Response to reviewer's comments

Response to reviewer 3

1. The author think that CNQDs is the proton transporter of the present system, what is the direct experiment or theoretical evidence to support this conclusion? Is it from EPR that CNQDs/CN has a stronger H radical signal than CN? If it is, why a strong H radical signal could be relevant to that CNQDs promote proton transport. If it is not, please give more detail that why CNQDs function as the proton transporter.

√ We acknowledge the reviewer's comments on the description of proton transport evidences. The reason why the CNQDs act as the proton transporter was proofed by the proton conductivity test with electrochemical AC impedance measurement under different relative humidity (RH), which directly reflects the capacity of proton transport on the surface of CNQDs, CNS, and CNQDs/CNS¹⁻⁴. As shown in Fig. 1i and supplementary Fig 16-19, it can be seen that the proton conductivities (σ) of CNQDs, CNS, and CNQDs/CNS are closely related to the relative humidity. The result implies that water molecules adsorbed on the surface of CNQDs, CNS, and CNQDs/CNS plays an important role in the proton transportation. The highest σ value (conductivity) was founded on the surface of CNQDs, no matter under any humidity condition, which can be attributed to the abundant strong polar functional groups on the CNQDs. These functional groups effectively adsorb water molecules to form a concentrated hydrogen bond network, allowing unobstructed proton transport pathways.

It also should be noticed that the conductivity of CNS depends on the relative humidity due the similar surficial functional groups. However, in terms of both quantity and quality, these functional groups on CNS are much less than that of CNQDs, so that the hydrogen bond network is unable to be constructed adequately, and resulting in a poor proton conductivity. Therefore, the σ value of CNQDs/CNS was 3.4 times higher than that of CNS under the 100% RH, which suggested that the modification of CNS by CNQDs intensively reinforce the proton transportation of CNS. We complement these discussions to the revised manuscript and the detailed description can be found in line 126-143, Page 6.

Fig. 1 i Proton conductivities (σ) of the CNQDs, CNS and CNQDs/CNS with respect to RH at 298 K.

Supplementary Figure 16. Nyquist plots of the CNQDs, CNS and CNQDs/CNS at 100% RH and 298 K.

Supplementary Figure 17. Nyquist plots of the CNQDs under 298 K conditions and different RH: **a** 100%, **b** 90%, **c** 70%, **d** 50%.

Supplementary Figure 18. Nyquist plots of the CNS under 298 K conditions and different RH: **a** 100%, **b** 90%, **c** 70%, **d** 50%.

Supplementary Figure 19. Nyquist plots of the CNQDs/CNS under 298 K conditions and different RH: **a** 100%, **b** 90%, **c** 70%, **d** 50%.

Reference

- 1 Huang, Y. G. *et al.* Selective CO₂ Capture and High Proton Conductivity of a Functional Star-of-David Catenane Metal–Organic Framework. *Adv. Mater* **29** (2017).
- 2 Wei, Y. S. *et al.* Unique Proton Dynamics in an Efficient MOF-Based Proton Conductor. *J. Am. Chem. Soc.* **139**, 3505–3512 (2017).
- 3 Zhai, Q.-G. *et al.* Cooperative Crystallization of Heterometallic Indium–Chromium Metal–Organic Polyhedra and Their Fast Proton Conductivity. *Angew. Chem. Int. Ed. Engl.* **54**, 7886–7890 (2015).

- 4 Karim, M. R. *et al.* Graphene Oxide Nanosheet with High Proton Conductivity. *J. Am. Chem. Soc.* **135**, 8097-8100, (2013).

2. I cannot understand with the data presented Figure 4b, how could the author come to the description that: In sharp contrast to the CNS, the CNQDs/CNS produces a significantly enhanced H-atom signal peak; and The Pt₁@CNQDs/CNS composite has a comparable H-atom signal peak to that of the CNQDs/CNS.

√ Thanks a lot for the valuable comments. It is very helpful for this manuscript. In the previous version, we compared a single one characteristic peak of •H radical at 3460–3480 G, not all of the 9 characteristic peaks. This might lead to misunderstanding and confusing. This time, we extracted all the characteristic peaks of •H radical by deconvolution of the raw spectrum to measure the change of the peak areas accurately (Fig. 4). Now, it can be clearly observed that the DMPO-H signal strength of CNQDs/CNS is much stronger than that of CNS and Pt₁@ CNQDs/CNS. We have modified the corresponding analysis results in the manuscript.

Fig. 4 In situ EPR on intermediates. **a** Experimental EPR signals of DMPO spin adducts generated under irradiation of a 300 W Xe lamp ($\lambda > 400$ nm) and **b** the deconvolution of DMPO-H spectrum according to the experimental signal of a-diagram recorded for the CNS, Pt₁@CNS, CNQDs/CNS and Pt₁@CNQDs/CNS.

3. From my observation, the intensity of Pt₁@CNQDs/CNS has decreased obviously in contrast to CNQDs/CNS as well, which is similar to that of Pt₁@CNS and CNS. This could be indicative that protons reduction still mainly occurs on Pt (conclusion obtained based on the author's analysis of Pt₁@CNS and CNS), instead of CNQDs (which is claimed by the author).

√ Thanks for the reviewer's valuable comments. It is our oversight to conduct an incomprehensive analysis of the data. As statement above, we have extracted the signal peaks of •H radical from Fig. 4a and deconvolute the data (as shown in Fig. 4b, Table 1).

Considering this, we complemented the corresponding analysis sections in the revised manuscript (line 222–235).

Revised manuscript:

“In order to better analyze the H-atoms transfer mechanism over different surfaces, the deconvolution of DMPO-H spectra was conducted using the Bruker Xepr software (Fig. 4b). The hyperfine coupling constants (hfcc) for nitrogen (a_N) and hydrogen (a_H) atoms were shown in Table 1, the hfcc values are well consistent with the previous reports^{54,55}, further confirming the formation of \bullet H radical. Moreover, the \bullet H radical signal strengths of the samples were also compared (the peak area in Table 1). It is found that the signal strength of CNQDs/CNS is stronger than that of CNS, indicating that the CNQDs can boost proton reduction to H-atoms.

In contrast, the $Pt_1@CNS$ results in a very low intensity of the H-atom, which is lower than that of the CNS. We propose that both the reduction of protons to H-atoms

and the recombination of H-atoms to H_2

which occurs on the Pt_1 sites⁵⁶. The signal strength of $Pt_1@CNQDs/CNS$ is lower than that of the CNQDs/CNS because Pt_1 accelerates the H-atom recombination reaction in aqueous solution⁵⁷

resulting in a decrease in the signal of H-atom. Based on the above analysis, it is reasonable to explain that the reduction of protons to H-atoms takes place in CNQDs in the $Pt_1@CNQDs/CNS$ system, and then the H-atoms are transferred to Pt_1 sites leading to the formation of H_2 . These results demonstrate that the Pt_1 sites serve as the H_2 bubble evolution outlets.”

Table 1 Relevant data from simulations of the EPR spectra of DMPO-H radical. The hyperfine coupling constants were obtained by fitting spectra using Bruker Xepr software.				
Samples	a_N/ G	a_H/ G	NoH=a_N/a_H	Peak area
CNS	16.29	22.54	0.72	1.15
$Pt_1@CNS$	16.31	22.62	0.72	0.39
CNQDs/CNS	16.30	22.52	0.72	1.95
$Pt_1@CNQDs/CNS$	16.28	22.59	0.72	0.81

4. Moreover, even if the above description is right and EPR test has been carried out to prove that the intensity of the signal is dose-related, I insist that these discrepancies could be a result of accidental error because they are so small. Repeated EPR should be carried out to prove the data reliability.

√ According to the reviewer's suggestion, the EPR experiment was re-conducted in another instrument, the results showed that the signal strength and the peak sharp are both similar with the results obtained previously. In fact, the EPR signal in Fig. 4a is derived from two adduct species of both hydrocarbon free radical and •H radical. It should be noticed that the signal intensity of hydrocarbon free radical is much higher than that of •H radical, and the signal peaks of the two radicals are partially overlapped. In other word, the peaks of hydrocarbon free covered the peaks of radical •H radical. This time, we use Bruker Xepr software to extract the signal of •H radical from the convolution data (Fig. 4b), expose a clear •H radical peaks. In addition, the hyperfine coupling constants (hfcc) is the key parameter to identify •H radical. As shown in Table 1, the hfcc values correspond well to previously reported values for the •H radical¹⁻⁴, which proves that the reliability of the data. The raw processing data is attached below (Fig.A-D). The red line is the original experimental data, and the yellow line is the fitting •H signal. It can be seen that the yellow line coincides with the original data very well. Thank you very much for your valuable suggestions and questions on our manuscript, which is very helpful for improving our manuscript.

Reference:

Table 1

Isotropic hyperfine coupling constants and NoH (N over H values) using DMPO. The isotropic hyperfine constants, NoH and literature values for the various DMPO adducts are given.

Adduct	a _N /Gauss	a _H /Gauss	NoH = a _N /a _H	Ref.
DMPO-CHO	15.72	21.27	0.74	this work
	15.80	21.10	0.75	[11]
DMPO-OH	14.94	14.88	1.004	this work
	14.90/15.00	14.90/15.00	1	[30,31]
DMPO-H	16.61	22.30	0.75	this work
	16.60	22.50	0.74	[30]
	16.60	22.60	0.73	[32]

Table from ref 3.

- Li, R. et al. Oxygen-controlled hydrogen evolution reaction: molecular oxygen promotes hydrogen production from formaldehyde solution using Ag/MgO nanocatalyst. *ACS Catal.* **7**, 1478-1484 (2017).

- 2 Zhao, L. M. et al. Photocatalysis with quantum dots and visible light: Selective and efficient oxidation of alcohols to carbonyl compounds through a radical relay process in water. *Angew. Chem. Int. Ed. Engl.* **56**, 3020-3024 (2017).
- 3 Bauer, N. A., Hoque, E., Wolf, M., Kleigrewe, K. & Hofmann, T. Detection of the formyl radical by EPR spin-trapping and mass spectrometry. *Free. Radic. Bio. Med.* **116**, 129-133 (2018).
- 4 Zhang, S., Yang, R., Zhao, W., Liang, Q. & Zhang, Z. The first ESR observation of radical species generated under pulsed electric fields processing. *LWT – Food. Sci. Technol.* **44**, 1233- 1235 (2011).

Fig. A The experimental EPR signals (red line) of DMPO spin adducts and the simulation DMPO-H spectrum (yellow line) of the CNS.

Fig. B The experimental EPR signals (red line) of DMPO spin adducts and the simulation DMPO-H spectrum (yellow line) of the Pt₁@CNS.

Fig. C The experimental EPR signals (red line) of DMPO spin adducts and the simulation DMPO-H spectrum (yellow line) of the CNQDs/CNS.

Fig. D The experimental EPR signals (red line) of DMPO spin adducts and the simulation DMPO-H spectrum (yellow line) of the $Pt_1@CNQDs/CNS$.

Decision letter and referee reports: third round

22nd Jun 20

Dear Prof Yin,

Your manuscript titled "Photocatalytic hydrogen production on Pt¹@CNQDs/CNS composites: Proton-assisted electron transfer and H-atom diffusion" has now been seen again by Reviewer 3, whose comment appears below. In light of their advice I am delighted to say that we are happy, in principle, to publish a suitably revised version in Communications Materials under the open access CC BY license (Creative Commons Attribution v4.0 International License).

We therefore invite you to edit your manuscript to comply with our format requirements and to maximise the accessibility and therefore the impact of your work.

EDITORIAL REQUESTS:

* Your manuscript should comply with our policies and format requirements, detailed in our checklist for authors at:

<https://www.nature.com/documents/commsmat-checklist.pdf>

* Please consider the following revised title and abstract, which have been edited for clarity. Please note that we do not allow abbreviations in titles, and defining "Pt1@CNQDs/CNS" in the title would take too many words. I therefore suggest simply describing this as a "model system" and full details are then given in the abstract.

Title: Proton-assisted electron transfer and hydrogen-atom diffusion in a model system for photocatalytic hydrogen production

Abstract: Solar energy can be converted into chemical energy by photocatalytic water splitting to produce molecular hydrogen. Details of the photo-induced reaction mechanism occurring on the surface of a semiconductor are not fully understood, however. Herein, we employ a model photocatalytic system consisting of single atoms deposited on quantum dots that are anchored on to a primary photocatalyst to explore fundamental aspects of photolytic hydrogen generation. Single platinum atoms (Pt1) are anchored onto carbon nitride quantum dots (CNQDs), which are loaded onto graphitic carbon nitride nanosheets (CNS), forming a Pt1@CNQDs/CNS composite. Pt1@CNQDs/CNS provides a well-defined photocatalytic system in which the electron and proton transfer processes that lead to the formation of hydrogen gas can be investigated. Results suggest that hydrogen bonding between hydrophilic surface groups of the CNQDs and interfacial water molecules facilitates both proton-assisted electron transfer and sorption/desorption pathways. Surface bound hydrogen atoms appear to diffuse from CNQDs surface sites to the deposited Pt1 catalytic sites leading to higher hydrogen-atom fugacity surrounding each isolated Pt1 island. We identify a pathway that allows for hydrogen-atom recombination into molecular hydrogen and eventually to hydrogen bubble evolution.

* I suggest changing the title of the Results section to 'Results and Discussion', and then removing final Discussion heading. The final paragraph of the paper ("A Pt1@CNQDs/CNS photocatalytic system with...") can then begin with "In summary,...".

* I suggest adding sub-headings throughout the Results and Discussion, the first of which should appear after the main 'Results and Discussion' heading.

* In the Author Contributions section please give initials rather than full names.

* The figure captions should begin with a short title in bold.

* Please remove the title page from the Supplementary Information.

* Communications Materials uses a transparent peer review system, where by we are publishing the reviewer comments to the authors, author rebuttal letters and journal decision letters of our research articles online as a supplementary peer review file. Please let us know in the cover letter when submitting the final version of your manuscript if you wish to opt in or opt out of transparent peer review. If you are concerned about the release of confidential data, we do permit redactions in the interest of confidentiality. If you would like to remove such information from these files, then please let us know specifically what information you would like to have removed. Please note that we cannot incorporate redactions for other reasons. Reviewer names will be published in the peer review files if the reviewer comments to the authors are signed by the reviewer, or if reviewers explicitly agree to release their name.

* Data availability statements and data citations policy: All Communications Materials manuscripts must include a section titled "Data Availability" at the end of the Methods section or main text (if no Methods). More information on this policy, and a list of examples, is available at http://www.nature.com/authors/policies/data/data-availability-statements-data-citations.pdf.

- Accession codes for deposited data
- Other unique identifiers (such as DOIs and hyperlinks for any other datasets)
- At a minimum, a statement confirming that all relevant data are available from the authors
- If applicable, a statement regarding data available with restrictions
- If a dataset has a Digital Object Identifier (DOI) as its unique identifier, we strongly encourage including this in the Reference list and citing the dataset in the Data Availability Statement.

DATA SOURCES: We strongly encourage authors to deposit all new data associated with the paper in a persistent repository where they can be freely and enduringly accessed. We recommend submitting the data to discipline-specific, community-recognized repositories, where possible and a list of recommended repositories is provided at http://www.nature.com/sdata/policies/repositories.

If a community resource is unavailable, data can be submitted to generalist repositories such as figshare or Dryad Digital Repository. Please provide a unique identifier for the data (for example a DOI or a permanent URL) in the data availability statement, if possible. If the repository does not provide identifiers, we encourage authors to supply the search terms that will return the data. For data that have been obtained from publically available sources, please provide a URL and the specific data product name in the data availability statement. Data with a DOI should be further cited in the methods reference section.

Please refer to our data policies at http://www.nature.com/authors/policies/availability.html.

* Please check whether your manuscript contains third-party images, such as figures from the literature, stock photos, clip art or commercial satellite and map data. We strongly discourage the use or adaptation of previously published images, but if this is unavoidable, please request the necessary rights documentation to re-use such material from the relevant copyright holders and

return this to us when you submit your revised manuscript.

* Your paper will be accompanied by a two-sentence editor's summary, of between 250-300 characters, when it is published on our homepage. Could you please approve the draft summary below or provide us with a suitably edited version.

"The chemical pathways by which photocatalytic hydrogen production occurs remain to be fully understood. Here, a model system is studied, composed of single atoms deposited on quantum dots, attached to a primary photocatalyst."

OPEN ACCESS:

Communications Materials is a fully open access journal. Articles are made freely accessible on publication under a [CC BY](http://creativecommons.org/licenses/by/4.0) license (Creative Commons Attribution 4.0 International License). This license allows maximum dissemination and re-use of open access materials and is preferred by many research funding bodies.

For further information about article processing charges, open access funding, and advice and support from Nature Research, please visit <https://www.nature.com/commsmat/about/open-access>

SUBMISSION INFORMATION:

In order to accept your paper, we require the following:

- * A cover letter describing your response to our editorial requests.
 - * The final version of your text as a Word or TeX/LaTeX file, with any tables prepared using the Table menu in Word or the table environment in TeX/LaTeX and using the 'track changes' feature in Word.
 - * Production-quality versions of all figures, supplied as separate files. Figures divided into parts should be labelled with a lowercase bold a, b, and so on. To ensure the swift processing of your paper please provide the highest quality, vector format, versions of your images (.ai, .eps, .psd) where available. Text and labelling should be in a separate layer to enable editing during the production process. If vector files are not available then please supply the figures in which ever format they were compiled in and not saved as flat .jpeg or .TIFF files. Any chemical structures or schemes contained within figures should additionally be supplied as separate ChemDraw (.cdx) files. If your artwork contains any photographic images, please ensure these are at least 300 dpi.
 - * The final version of the Supplementary Information (figures, tables, notes etc) in one PDF file. Please submit movies, audio files and data sets as separate files. See <https://www.nature.com/commsmat/submit/submission-guidelines#supplementary-info> for acceptable file formats/sizes.
- ** Please note that Supplementary Information cannot be changed after the paper has been accepted **
- * An updated <https://www.nature.com/documents/nr-editorial-policy-checklist.pdf> Editorial Policy checklist, uploaded as a 'Related Manuscript File' type. This checklist is to ensure your paper complies with all relevant editorial policies. Please note that this form is a dynamic 'smart pdf' and must therefore be downloaded and completed in Adobe Reader, instead of opening it in a web browser.

* If you wish, an interesting image (but not an illustration or schematic) for consideration as the banner image on our homepage. The file should be 1400x400 pixels in RGB format and should be uploaded as 'Related Manuscript File'. In addition to our home page, we may also use this image (with credit) in other journal-specific promotional material.

At acceptance, the corresponding author will be required to complete an Open Access Licence to Publish on behalf of all authors, declare that all required third party permissions have been obtained and provide billing information in order to pay the article-processing charge (APC) via credit card or invoice.

Please note that your paper cannot be sent for typesetting to our production team until we have received these pieces of information; therefore, please ensure that you have this information ready when submitting the final version of your manuscript.

ORCID

Communications Materials is committed to improving transparency in authorship. As part of our efforts in this direction, we are now requesting that all authors identified as 'corresponding author' create and link their Open Researcher and Contributor Identifier (ORCID) with their account on the Manuscript Tracking System (MTS) prior to acceptance. ORCID helps the scientific community achieve unambiguous attribution of all scholarly contributions. For more information please visit <http://www.springernature.com/orcid>

For all corresponding authors listed on the manuscript, please follow the instructions in the link below to link your ORCID to your account on our MTS before submitting the final version of the manuscript. If you do not yet have an ORCID you will be able to create one in minutes.

IMPORTANT: All authors identified as 'corresponding author' on the manuscript must follow these instructions. Non-corresponding authors do not have to link their ORCIDs but are encouraged to do so. Please note that it will not be possible to add/modify ORCIDs at proof. Thus, if they wish to have their ORCID added to the paper they must also follow the above procedure prior to acceptance.

To support ORCID's aims, we only allow a single ORCID identifier to be attached to one account. If you have any issues attaching an ORCID identifier to your MTS account, please contact the [Platform Support Helpdesk](http://platformsupport.nature.com/).

[link redacted]

We hope to hear from you within two weeks; please let us know if the process may take longer.

Best regards,

John Plummer, PhD
Chief Editor
orcid.org/0000-0003-4824-8497

Communications Materials

REVIEWERS' COMMENTS:

Reviewer #3 (Remarks to the Author):

The authors have addressed the questions properly and it now could be accepted for publication.